# Dynamic prediction model of spontaneous combustion risk in goaf based on improved CRITIC-G2-TOPSIS method and its application

Wei Wang[1,2,3]*, Yun Qi[1,2,3]*, Baoshan Jia[2,3], Youli Yao[1]

**1** School of Coal Engineering, Shanxi Datong University, Datong, P.R. China, **2** College of Safety Science and Engineering, Liaoning Technical University, Fuxin, P.R. China, **3** Key Laboratory of Mine Thermodynamic disasters and Control of Ministry of Education (Liaoning Technical University), Huludao, P.R. China

* wangwei@sxdtdx.edu.cn (WW); qiyun_sx@sxdtdx.edu.cn (YQ)

**Data Availability Statement:** All relevant data are within the paper and its Supporting Information files.

## Abstract

Due to the problems related to the numerous factors affecting the spontaneous combustion of goaf coal, such as sudden, uncertain, and dynamic changes, and the fact that the weight of the indexes in the prediction model of the spontaneous combustion risk is difficult to determine, an improved Criteria Importance Through Inter-criteria Correlation (CRITIC) modified Technique for Order of Preference by Similarity to Ideal Solution G2-(TOPSIS) dynamic prediction model of goaf spontaneous combustion was developed. An optimal decision-making model was established by introducing the Euclidean distance function, and the function-driven type G2 weighting method was modified using the differential-driven type weighting method of the CRITIC. In addition, the comprehensive weights of each index were obtained. An update factor was introduced to obtain the dynamic weight, and the primary-secondary relationship of the risk factors affecting the spontaneous combustion of goaf was evaluated. Based on the G2 weighting method, which approximates the driving function principle of the ideal solution ranking method (TOPSIS), a G2-TOPSIS goaf spontaneous combustion risk assessment model was established. The degree of closeness was analyzed and the risk grade of the goaf spontaneous combustion was finally predicted. The sub-model was applied to the goaf of working face 1303 in the Jinniu Coal Mine. It was concluded that the air leakage duration was the dominant factor inducing the risk of the spontaneous combustion of the goaf. The risk grade of spontaneous combustion of the goaf is Class III, and the predicted results are consistent with the actual situation. The improved CRITIC-G2-TOPSIS dynamic model was demonstrated to be scientific in predicting the goaf spontaneous combustion risk, and these research results have important popularization and application value.

## 1 Introduction

Spontaneous combustion of residual coal in goaf is a fire hazard and is affected by many factors [1,2]. As the mining depth and mining intensity increase, the initial temperature of the

**Funding:** Key special-funded projects of the State key R & D program (2018YFC0807900).

**Competing interests:** The authors have declared that no competing interests exist.

**Abbreviations:** AHP, Analytic Hierarchy Process; BP, back-propagation; CRITIC, Criteria Importance Through Inter-criteria Correlation; TOPSIS, Order of Preference by Similarity to Ideal Solution.

coal seam increases, especially in gassy and flammable coal seams, which increases the risk of spontaneous combustion of the goaf. However, for most cases, the spontaneous combustion occurred in the deep part of the coal mine, far from the working face. Furthermore, for such spontaneous combustion cases, only smoke was visible, making the fire difficult to locate [3,4]. Once spontaneous combustion occurs in a coal mine, the working face will become partially or fully blocked. Worse yet, disasters, such as gas explosions and dust explosions, are likely to follow [5]. Therefore, it is important and urgent to explore spontaneous goaf combustion control. In such research, the effective and immediate prediction of spontaneous combustion risks plays a crucial role.

Many scholars in China and abroad have already conducted a great deal of research from different perspectives in terms of the prediction of spontaneous combustion risks [6]. Evaluation methods, including fuzzy comprehensive evaluation, hierarchical analysis, the matter-element model, support vector machines, artificial neural networks, and the gray model are widely used in this field [7,8]. Researchers, including Singh et al. [9], have developed an expert system for evaluating the risks of spontaneous combustion in longwall mining faces. Padhee et al. [10] used a neural network model and a fuzzy expert system to predict the spontaneous combustion tendency. In addition, authors have proposed the use of the fuzzy C average method to classify coal spontaneous combustion risks. Another group of scholars, including Sizong [11], used fuzzy mathematics theory to study a method for the comprehensive prediction of spontaneous combustion risk in mining coal seams. At the end of the 20th century, researchers, including Huang Mengtao [12], first used neural networks to predict spontaneous combustion in coal seams, and a model for an experimental low-temperature spontaneous coal combustion furnace was then established. Hongfei [13] established a coal seam spontaneous combustion risk prediction model, which is an upgraded version of a back-propagation (BP) neural network, and carried out a case study analysis. Similarly, Gao [14] used the Analytic Hierarchy Process (AHP) to determine the weighting coefficients of all of the factors influencing the spontaneous combustion of goaf in the Wuda Mining Area and sorted them. Jun et al. [15] employed system engineering theory, the AHP method, and the fuzzy mathematics method to construct a comprehensive evaluation model of coal spontaneous combustion risks, and they identified five judgment indicators to evaluate the risk level of spontaneous combustion. Based on a horizontal comparison, Chaoyu [16] used the AHP method and the Technique for Order of Preference by Similarity to Ideal Solution (TOPSIS) method to establish a goaf spontaneous combustion risk analysis model and to compare the spontaneous coal combustion risks in mines. To some extent, these studies have played an important role in the prediction and prevention of spontaneous combustion. However, it is difficult to determine the weights of some of the evaluation indicators in actual applications of the evaluation model. In addition, the main and secondary factors affecting spontaneous coal combustion are not clear enough. Furthermore, the study of spontaneous goaf combustion risks is based on a static prediction model, but spontaneous goaf combustion is a complex, dynamic process with abruptness and uncertainty. Therefore, it is necessary to establish a dynamic prediction model to evaluate the possibility of spontaneous combustion.

In view of this, in this study, the goaf spontaneous combustion of working face 1303 in the Jinniu Coal Mine was taken as the research background, the improved CRITIC method, G2 method and TOPSIS method are combined to dynamic predict the risk of spontaneous combustion of residual coal in goaf. There are many studies on the application of CRITIC method [17,18] and TOPSIS method [19,20]. Iovi M et al. [21] proposed a CRITIC-M method. By introducing a new standardization process, a small deviation between standardized elements is realized, so that a lower standard deviation value can be obtained and the

relationship between data in the initial decision matrix can be presented in a more objective way. Liu et al. [22] using the improved AHP method and critical method are used to obtain the subjective weight and objective weight of each index respectively, and the comprehensive weight is obtained by fusion according to the principle of minimum identification information. The model can describe various uncertainties of interval value index and quickly and effectively determine the grade of rock burst intensity. Hui et al. [23] using the objective weight of the evaluation index is calculated by critical method, and the spontaneous combustion tendency of oil tank corrosion products is comprehensively evaluated by TOPSIS theory, so as to realize the accurate evaluation of the spontaneous combustion tendency of oil tank corrosion products.

The author makes a new improvement on critical method, according to the variation degree of each evaluation index and its influence on other indexes, the mean difference of evaluation indexes is introduced, and the ratio of information is used to replace the ratio of relative importance determined by decision-makers, so as to determine the index weight. G2 method can better reflect experts' knowledge, experience, information and risk awareness. In order to overcome the shortcomings of G2 method, the improved critical method is integrated into G2 method for coupling. With the help of the improved critical method, the importance ratio between indicators is corrected, so that the subjective weighting results have both data objectivity. G2 weighting method is based on the improved Criteria Importance Through Inter-criteria Correlation (CRITIC) information quantity. Owing to the variability and the contradictory nature of the index data, the disadvantages of the single weighting method were avoided. The single weighting method cannot reflect both the subjective and objective information. Targeting the dynamic changes in the goaf environment, weight renewal factors are introduced in the new method. Based on the TOPSIS method, the G2-TOPSIS dynamic prediction model of spontaneous goaf combustion was established to provide an objective evaluation of spontaneous goaf combustion risks. The results of this study lay a solid foundation for developing scientific and reasonable prevention measures for spontaneous goaf combustion and provide theoretical support for solutions to fire problems in other coal mines.

## 2 Improving CRITIC and revising the G2 weighting method

### 2.1 Improvement of the CRITIC method to determine the weights of the objective indicators

The improved CRITIC weighting method is a difference-driven objective weighting method, which is based on the variations in each evaluation index and the influences of the other indexes [24–26]. It introduces the mean deviation in the evaluation index. In terms of the determination of the index weight, it uses the information amount ratio instead of the decision-maker's subjective conclusion. The main calculation steps are as follows.

1) Index information amount

$$C_k = \frac{\sigma_k}{u_k} \sum_{i=1}^{m} (1 - |t_{ik}|), k = 1, 2, \ldots, m, \tag{1}$$

where $C_k$ is the improved CRITIC information quantity of the k[th] evaluation index, $\sigma_k$ is the standard deviation in the k[th] evaluation index, and $u_k$ is the average value of the k[th] evaluation index. $\sum_{i=1}^{m} (1 - |t_{ik}|)$ is the quantitative value of the interaction degree between the k[th] index and the other indicators, and $t_{ik}$ is the correlation coefficient for the evaluation indexes i and k.

2) Determination of the index weight

$$w_{ck} = \frac{C_k}{\sum\limits_{i=1}^{m} C_i}, j = 1, 2, 3, \ldots, m. \tag{2}$$

## 2.2 Determination of the G2 subjective index weight

The G2 weighting method is a function-driven subjective weighting method [27], which provides a scoring range for the evaluation index according to the relative importance of each evaluation index when the information and risks suffer from large uncertainties. It calculates the weight of each evaluation index in the group decision-making process through quantitative calculations. The calculation steps are as follows.

Based on the experts' own experiences and knowledge and considering their preference coefficients, the m indexes in the original index set $\{u_i\}$ are reordered as $\{u_{i1}, \ldots, u_{ik}, \ldots, u_{im}\}$ according to their importance. Among them, $u_{i1}$ is the most important index, and $u_{im}$ is the least important index. The experts have made rational judgments on the ratio $r_{km}$ of the importance of the evaluation index $u_{ik}$ to $u_{im}$.

$$r_{km} = a_k, k = 1, 2, \ldots, m - 1. \tag{3}$$

However, in special cases, because of the insufficiency of some of the information, the experts are not able to assign $a_k$ an exact value. Since they cannot assign it an exact value, they have to assign it a range of values $[d_{1k}, d_{2k}]$. The obtained bounded, closed, real number interval is an interval number that cannot be regarded and is composed of $d_1$ and $d_2$, which is denoted as $D_k$, where $D_k = [d_{1k}, d_{2k}]$.

$$e(D_k) = d_{2k} - d_{1k.}; \tag{4}$$

$$n(D_k) = \frac{1}{2}(d_{2k} + d_{1k.}); \tag{5}$$

$$\phi\varepsilon(D_k) = n(D_k) + \varepsilon e(D_k); \tag{6}$$

$$r_{km} = a_k \in [d_{1k}, d_{2k}], k = 1, 2, \ldots, m - 1. \tag{7}$$

In Eqs (4–7), $e(D_k)$ is the interval length; $n(D_k)$ is the interval midpoint; and $\phi\varepsilon(D_k)$ is the interval mapping function of the expert's risk attitude. $\varepsilon$ is the risk attitude factor ($|\varepsilon| \leq 0.5$). When $-0.5 \leq \varepsilon \leq 0$, it is conservative; when $\varepsilon = 0$, it is neutral; and when $0 \leq \varepsilon \leq 0.5$, it is a risk. For a certain expert, $\varepsilon$ is the deterministic number.

If the assignment of $\{D_k\}$ is accurate, $w_k$ (the k[th] index in the G2 method) is calculated as follows:

$$w_{Gk} = \phi\varepsilon(D_k) / \sum_{i=1}^{m} \phi\varepsilon(D_i), k = 1, 2, \ldots, m. \tag{8}$$

## 2.3 Determining the comprehensive weights for the improved G2 method (based on the improved CRITIC)

Considering the decision-makers' intuitive understanding of the rules and factors controlling spontaneous goaf combustion, which are reflected by objective survey data, $C_k$ (based on the improved CRITIC), the confidence interval of the observation values of the information quantity is recorded as $[C_{1k}, C_{2k}]$, where k = 1, 2,. . ., m. Then, the ratio of the upper

bounds to the lower bounds of the confidence interval of the improved CRITIC information is used to calculate the rational assignment interval artificially given in the interval proxy of the G2 method for the ratio of the degrees of the importance of the two indicators as follows:

$$R_{km} = \begin{cases} [d_{1k}, d_{2k}] = [\min(\frac{C_{1k}}{C_{1m}}, \frac{C_{2k}}{C_{2m}}), \max(\frac{C_{1k}}{C_{1m}}, \frac{C_{2k}}{C_{2m}})], C_k > C_m, k = 1, 2, \ldots, m \\ [1, 1], C_k \leq C_m \end{cases}. \tag{9}$$

$w$ (the k$^{\text{th}}$ evaluation index interval's assigned weight) is calculated as follows:

$$w = \frac{\frac{1}{2}(d_{1k} + d_{2k}) + \varepsilon(d_{2k} - d_{1k})}{\sum_{i=1}^{m} \frac{1}{2}(d_{1i} + d_{2i}) + \sum_{i=1}^{m} \varepsilon(d_{2i} - d_{1i})}, k = 1, 2, \ldots, m. \tag{10}$$

## 2.4 Determination of the dynamic weight

The dynamic weight is an improvement on the fixed weight in the traditional evaluation method [28]. It is the coupling of the renewal index factors based on the comprehensive weight of the improved CRITIC and the G2 method. It can revise the weight according to the feedback from each index in the evaluation system, which changes with the environment of the goaf, and can achieve the timely adjustment of the index to the hazard shadow of spontaneous goaf combustion, thus improving the flexibility of the evaluation system.

$$w_s(0) = \frac{w \cdot q(0)}{\sum_{k=1}^{n} w \cdot q(0)}; \tag{11}$$

$$\varphi(j) = 1 + |\Delta q|; \tag{12}$$

$$w_D = \frac{w_s(0) \cdot \varphi(j)}{\sum_{k=1}^{n} w_s(0) \cdot \varphi(j)}. \tag{13}$$

In Eqs (11–13), $w_s(0)$ is the standard weight of each index; $w$ is the comprehensive weight; $q(0)$ is the initial value of the experts' scoring of the spontaneous combustion index; $\Delta q$ is the absolute value of the difference between the scoring value of each index and its initial value at a certain time; $\varphi(j)$ is the updating factor of the weight; and $w_D$ is the dynamic weight.

## 3 G2-TOPSIS spontaneous goaf combustion dynamic evaluation model

### 3.1 Constructing the initial evaluation matrix

The TOPSIS method normalizes the original data matrix and determines the distances between the evaluation object and the positive ideal solution and the negative ideal solution using the cosine method [29,30]. In this way, it obtains the relative closeness between the evaluation object and the positive ideal solution [31]. Therefore, it can objectively reflect the actual situation.

1) If the multi-factor evaluation object set is $A = \{A_1, A_2, \ldots, A_m\}$, the measured quality of the object is $R_1, R_2, \ldots, R_n$, and the evaluation objects in A, i.e., the attribute set of the evaluation index, is $A_i$ ($i = 1, 2, \ldots m$), then their vectors $[a_{i1}, a_{i2}, \ldots, a_{in}]$, which are composed of $n$ attribute values of the index, can uniquely represent the object $A_i$ as a point in n-dimensional space. In addition, the evaluation index $a_{ij}$ represents the $j^{\text{th}}$ attribute value of the $i^{\text{th}}$ evaluation object; $i \in [1, m]$ and $j \in [1, n]$; so, the initial evaluation matrix is

$$A = (a_{ij})_{m \times n} = \begin{bmatrix} a_{11} & a_{12} & \cdots & a_{1n} \\ a_{21} & a_{22} & \cdots & a_{2n} \\ \vdots & \vdots & \ddots & \vdots \\ a_{m1} & a_{m2} & \cdots & a_{mn} \end{bmatrix}. \tag{14}$$

2) Because the dimensions of the indexes are different, it is necessary to normalize the attribute values of the indexes and transform the values into the interval of [0,1]. The normalized evaluation matrix is $B_{ij} = (b_{ij})_{m \times n}$, in which

$$b_{ij} = a_{ij} / \sqrt{\sum_{i=1}^{m} a_{ij}^2}. \tag{15}$$

3) The weighted evaluation matrix $Z_{ij}$ is constructed, and the weight matrix $w$ of the evaluation index is multiplied by the normalized evaluation matrix $B_{ij}$, which is obtained using the TOPSIS method, using the improved CRITIC and revised G2 methods to form a weighted comprehensive evaluation matrix:

$$Z_{ij} = B_{ij} \cdot w = \begin{bmatrix} f_{11} & f_{12} & \cdots & f_{1n} \\ f_{21} & f_{22} & \cdots & f_{2n} \\ \vdots & \vdots & \ddots & \vdots \\ f_{m1} & f_{m2} & \cdots & f_{mn} \end{bmatrix}. \tag{16}$$

## 3.2 Determining the computational similarities of the positive and negative ideal solutions

1) The positive and negative ideal solutions of the evaluation objectives are obtained according to the weighted comprehensive evaluation matrix:

$$f^+ = \{(\max b_{ij} | j \in J^+), (\min b_{ij} | j \in J^-)\}; \tag{17}$$

$$f^- = \{(\min b_{ij} | j \in J^+), (\max b_{ij} | j \in J^-)\},$$

where $J^+$ is the benefit index, and $J^-$ is the cost index.

2) The Euclidean distance between the objective value and the ideal solution of each evaluation is

$$S_i^+ = \sqrt{\sum_{j=1}^m (f_{ij} - f_j^+)^2}, \quad i = 1, 2, \cdots, n;$$

(18)

$$S_i^- = \sqrt{\sum_{j=1}^m (f_{ij} - f_j^-)^2}, \quad i = 1, 2, \cdots, n.$$

3) The relative closeness between each evaluation object and the optimal value is calculated as follows:

$$N_i^+ = S_i^- / (S_i^+ + S_i^-).$$

(19)

For $N_i^+$, the larger the value, the closer the evaluation object to the ideal solution and the better the evaluation object. Based on the relative degree of closeness, the evaluation objects are ranked, and a decision-making basis is formed.

### 3.3 Constructing the G2-TOPSIS dynamic evaluation model

According to the principle of combination weighting, the means of the dynamic weights of the criteria level indicators, $\overline{w}_D$, and the expert evaluation matrix, $V$, based on the CRITIC and G2 improved method after reordering the alignment of the side layer indicators, the comprehensive evaluation result $L$ is calculated as follows:

$$L = \overline{w}_D \cdot V.$$

(20)

The G2-TOPSIS comprehensive evaluation model quantifies the qualitative criteria for spontaneous goaf combustion. In addition, it introduces updated factors to modify the index weights, obtaining dynamic weights. The spontaneous goaf combustion risk grade is evaluated using the dynamic prediction model of spontaneous goaf combustion based on the improved CRITIC and G2-TOPSIS method. The calculation process is shown in Fig 1.

## 4 Construction of the spontaneous goaf combustion risk prediction system

### 4.1 Spontaneous Goaf combustion risk analysis

A large amount of heat, accumulated by the oxidation of the residual coal in the goaf, is the main factor leading to spontaneous combustion [32]. Considering the oxidation of the residual coal and the factors related to the heat generation and heat dissipation [33], the factors influencing the spontaneous goaf combustion can be divided into four aspects: the spontaneous coal combustion tendency, the air leakage oxygen supply conditions, the heat storage and heat dissipation conditions, and the goaf conditions and specifications. The influence of the propensity for spontaneous coal combustion on the spontaneous combustion risk was analyzed and divided into the degree of carbonization and metamorphism, the rate of increase of the CO unit temperature, the temperature difference of the coal oxidation and reduction ignition point, and the oxygen absorption capacity of the coal. The lithology, fracture development,

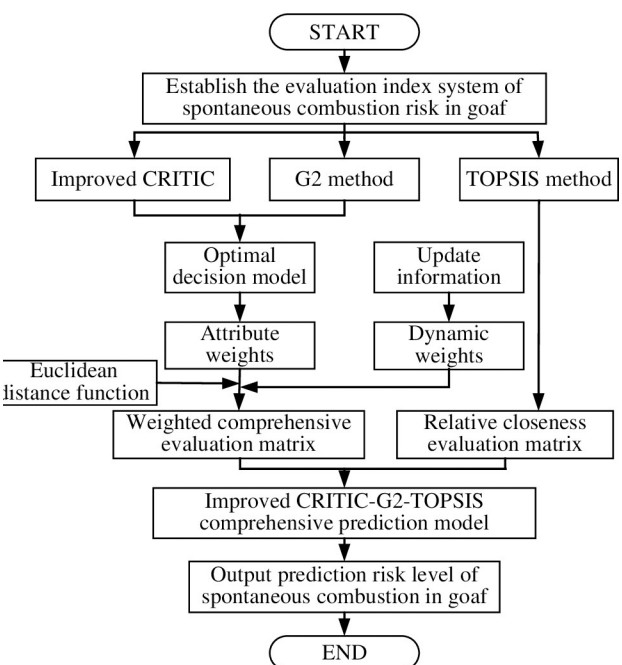

**Fig 1. Flowchart of the G2-TOPSIS dynamic evaluation model.**

geological structure, and air supply conditions of the goaf roof and floor are the important factors related to the air leakage and oxygen supply conditions in the goaf. The impacts can be divided into the coal seam thickness, the depth of the coal seam, the geological structure, and the dip angle of the coal seam. The key factor affecting spontaneous coal combustion is that the heat accumulated by the oxidation of the residual coal in the goaf reaches the burning point of coal in unit time. The heat storage and heat dissipation conditions were analyzed and divided into the air leakage intensity, the temperature of the surrounding rock, and the air leakage time. The status and specifications of the goaf reflect its dynamic changes during working face mining, which can be divided into the residual coal thickness, the advance speed, the goaf area, and the air supply. When determining the index weight, dynamic weight is adopted for 15 secondary indicators.

In order to facilitate the evaluation of the spontaneous goaf combustion risk, referring to the relevant literature [34], a prediction system for these factors was established (Table 1). In this system, the spontaneous goaf combustion risk is the target layer and the single index is the criterion layer.

## 4.2 Establishing the evaluation levels of the spontaneous goaf combustion risks

To quantitatively study the spontaneous goaf combustion risks, the Delphi method has been used by experts and technicians to classify and assign spontaneous combustion risks. According to the characteristics of the safety production of goaf, the spontaneous combustion risks were divided into five levels: I (basically impossible), II (less likely to occur), III (likely to occur), IV (pretty likely to occur), and V (very likely to occur) (Table 2). Each level is assigned within [0,100]. The critical value of each grade is determined by the relevant regulations [35]. The higher the assigned value of each index, the smaller the spontaneous goaf combustion risk, and the smaller the possibility of spontaneous goaf combustion.

Table 1. Prediction and evaluation system for spontaneous combustion hazards.

| Spontaneous goaf combustion risks $U_i$ | Single index |
|---|---|
| spontaneous combustion tendency $U_1$ | degree of coal metamorphism $U_{11}$ |
| | rate of CO temperature increase per unit $U_{12}$ |
| | temperature difference of coal oxidation-reduction ignition point $U_{13}$ |
| | coal oxygen inhalation capability $U_{14}$ |
| air leakage and oxygen supply conditions $U_2$ | thickness of coal seam $U_{21}$ |
| | depth of coal seam $U_{22}$ |
| | geological structure $U_{23}$ |
| | slope of coal seam $U_{24}$ |
| heat accumulation and emission conditions $U_3$ | air leakage strength $U_{31}$ |
| | temperature of surrounding rock $U_{32}$ |
| | air leakage time $U_{33}$ |
| goaf conditions and specifications $U_4$ | residual coal thickness $U_{41}$ |
| | advance speed $U_{42}$ |
| | goaf area $U_{43}$ |
| | air supply $U_{44}$ |

# 5 Application example of spontaneous goaf combustion risk prediction model

## 5.1 Establishing the goaf spontaneous combustion risk prediction object and data

Taking the Shanxi Jinniu Coal Mine as an example, fully mechanized caving face 1303 was analyzed using the improved CRITIC-G2-TOPSIS dynamic prediction model. Working face 1303 is located at level 1030 in mining coal seam 9. The coal seam is 6.17-m thick and has a slope of 8˚–14˚ (average of 10˚), making it a grade II spontaneous combustion coal seam. Its coal dust explosion index is 45.79%, i.e., explosive.

The degree of the spontaneous goaf combustion risk of the working face was scored three times by experts, and the scoring intervals lasted for several days. The CRITIC method was used to re-rank the single index of the spontaneous combustion risk. Experts were invited to grade the 15 single indexes after re-ranking. Because the experts could not provide the exact figures of the risk assess capability, an interval number was used to replace the probability of spontaneous combustion caused by each index, and the exact number was used to evaluate every single index (Table 3).

## 5.2 Establishing the indicator weights and the weighted judgment matrix

According to Table 3, the weights of each evaluation index were calculated using the G2 weighting method (modified using CRITIC). The risk attitude factor of the experts was $\varepsilon$ = 0.25. Therefore, the index scores of the criterion level can be substituted into Eq (10) to obtain the weight matrix $w$. The results are shown in Table 4.

Table 2. Evaluation grades and value ranges for spontaneous goaf combustion.

| Grade | I | II | III | IV | V |
|---|---|---|---|---|---|
| Value range | (90,100] | (70,90] | (60,70] | (40,60] | (0,40] |

**Table 3. Expert scores and improved indicator scores.**

| Single index | Expert evaluation | First evaluation | Second evaluation | Third evaluation |
|---|---|---|---|---|
| degree of coal metamorphism $U'11$ | (0.4,0.6) | 70 | 80 | 60 |
| rate of increase of CO temperature $U'12$ | (0.8,1.0) | 50 | 70 | 50 |
| temperature difference of coal oxidation-reduction ignition point $U'13$ | (0.7,0.9) | 60 | 80 | 70 |
| coal oxygen absorption capacity $U'14$ | (0.6,0.8) | 50 | 60 | 60 |
| thickness of coal seam $U'21$ | (0.5,0.7) | 30 | 40 | 30 |
| depth of coal seam $U'22$ | (0.6,0.8) | 50 | 90 | 80 |
| geological structure $U'23$ | (0.3,0.5) | 60 | 80 | 70 |
| slope of coal seam $U'24$ | (0.4,0.6) | 70 | 90 | 80 |
| air leakage strength $U'31$ | (0.7,1.0) | 60 | 80 | 70 |
| temperature of surrounding rock $U'32$ | (0.7,0.9) | 50 | 80 | 60 |
| air leakage time $U'33$ | (0.7,0.9) | 80 | 70 | 80 |
| residual coal thickness $U'41$ | (0.3,0.5) | 50 | 60 | 55 |
| advance speed $U'42$ | (0.7,0.8) | 60 | 80 | 70 |
| goaf area $U'43$ | (0.4,0.6) | 60 | 60 | 80 |
| air supply $U'44$ | (0.5,0.7) | 60 | 70 | 65 |

Considering the evaluation score from the experts on the dynamic changes in the spontaneous combustion information on the goaf, the engineering attribute weight, weight updating factor, and dynamic weight of the index were calculated using Eqs (11–13). The calculation results are shown in Table 5. As can be seen, as the working face is advanced, the weight of each spontaneous goaf combustion index changes. The main factors causing spontaneous combustion in goaf are the length of the air leakage, the temperature of the surrounding rock, the temperature difference of the coal oxidation and reduction ignition point, the air supply rate, and the oxygen absorption capacity of the coal.

After introducing the weight updating factors, the change in the index attribute weight is shown in Fig 2. The analysis shows that the index attribute weight changes dynamically as the goaf changes, and the dynamic weight can adequately describe the change process. Among them, the weights of the temperature of the surrounding rock, the air leakage time, the air leakage intensity, the coal seam thickness, and the air supply volume are greatly affected by the environmental changes in the goaf.

Matrix **A** was established according to the expert's score for each criterion level index.

$$A = \begin{bmatrix} 70 & 50 & 60 & 50 & 30 & 50 & 60 & 70 & 60 & 50 & 80 & 50 & 60 & 60 & 60 \\ 80 & 70 & 80 & 60 & 40 & 90 & 80 & 90 & 80 & 80 & 70 & 60 & 80 & 60 & 70 \\ 60 & 50 & 70 & 60 & 30 & 80 & 70 & 80 & 70 & 60 & 80 & 55 & 70 & 80 & 65 \end{bmatrix}.$$

**Table 4. Weights of the evaluation indexes.**

| Index | $U'11$ | $U'12$ | $U'13$ | $U'14$ | $U'21$ | $U'22$ | $U'23$ | $U'24$ |
|---|---|---|---|---|---|---|---|---|
| Weight | 0.052 | 0.090 | 0.081 | 0.071 | 0.062 | 0.071 | 0.043 | 0.052 |
| Index | $U'31$ | $U'32$ | $U'33$ | $U'41$ | $U'42$ | $U'43$ | $U'44$ | --- |
| Weight | 0.088 | 0.081 | 0.081 | 0.043 | 0.073 | 0.052 | 0.062 | --- |

$w = diag(0.052\ 0.090\ 0.081\ 0.072\ 0.062\ 0.071\ 0.043\ 0.052\ 0.088\ 0.081\ 0.043\ 0.073\ 0.052\ 0.062)$.

**Table 5. Update indicators and dynamic weights of each indicator.**

| Index | Dynamic evaluation | | | $\Delta_q$ | | | Renewed factors $\varphi(j)$ | | | Standard weight | | | Dynamic weight | | | Average weight |
|---|---|---|---|---|---|---|---|---|---|---|---|---|---|---|---|---|
| | 1 | 2 | 3 | 1 | 2 | 3 | 1 | 2 | 3 | 1 | 2 | 3 | 1 | 2 | 3 | |
| $U'11$ | 74 | 80 | 60 | 4 | 0 | 0 | 5 | 1 | 1 | 0.0635 | 0.0568 | 0.0474 | 0.123 | 0.027 | 0.0203 | 0.0568 |
| $U'12$ | 53 | 70 | 50 | 3 | 0 | 0 | 4 | 1 | 1 | 0.0785 | 0.086 | 0.0684 | 01216 | 0.041 | 0.0293 | 0.064 |
| $U'13$ | 62 | 78 | 68 | 2 | 2 | 2 | 3 | 3 | 3 | 0.0847 | 0.0885 | 0.0837 | 0.0984 | 0.126 | 0.1077 | 0.1107 |
| $U'14$ | 50 | 62 | 63 | 0 | 2 | 3 | 1 | 3 | 4 | 0.0619 | 0.0581 | 0.068 | 0.024 | 0.083 | 0.1167 | 0.0746 |
| $U'21$ | 30 | 41 | 32 | 0 | 1 | 2 | 1 | 2 | 3 | 0.0324 | 0.0339 | 0.0301 | 0.0125 | 0.0323 | 0.0387 | 0.0278 |
| $U'22$ | 50 | 89 | 80 | 0 | 1 | 0 | 1 | 2 | 1 | 0.0619 | 0.0873 | 0.0863 | 0.024 | 0.0831 | 0.037 | 0.048 |
| $U'23$ | 62 | 81 | 69 | 2 | 1 | 1 | 3 | 2 | 2 | 0.045 | 0.047 | 0.0451 | 0.0523 | 0.0447 | 0.0387 | 0.0452 |
| $U'24$ | 71 | 90 | 81 | 1 | 0 | 1 | 2 | 1 | 2 | 0.0635 | 0.0639 | 0.064 | 0.0271 | 0.0304 | 0.0549 | 0.0375 |
| $U'31$ | 60 | 80 | 70 | 0 | 0 | 0 | 1 | 1 | 1 | 0.0921 | 0.0961 | 0.0936 | 0.0357 | 0.0457 | 0.0402 | 0.0405 |
| $U'32$ | 52 | 82 | 63 | 2 | 2 | 3 | 3 | 3 | 4 | 0.0706 | 0.0885 | 0.0775 | 0.082 | 0.1263 | 0.133 | 0.1138 |
| $U'33$ | 83 | 73 | 82 | 3 | 3 | 2 | 4 | 4 | 3 | 0.113 | 0.0774 | 0.1009 | 0.1751 | 0.1473 | 0.13 | 0.1508 |
| $U'41$ | 48 | 62 | 54 | 2 | 2 | 1 | 3 | 3 | 2 | 0.0375 | 0.0352 | 0.0353 | 0.0436 | 0.0502 | 0.0303 | 0.0414 |
| $U'42$ | 59 | 80 | 71 | 1 | 0 | 1 | 2 | 1 | 2 | 0.0764 | 0.0797 | 0.0787 | 0.0592 | 0.0379 | 0.0675 | 0.0549 |
| $U'43$ | 60 | 61 | 78 | 0 | 1 | 2 | 1 | 2 | 3 | 0.0544 | 0.0425 | 0.0616 | 0.0211 | 0.0404 | 0.0793 | 0.047 |
| $U'44$ | 63 | 68 | 63 | 3 | 2 | 2 | 4 | 3 | 3 | 0.0649 | 0.0592 | 0.0593 | 0.1005 | 0.0845 | 0.0763 | 0.0871 |

Eq (15) was used to normalize and obtain matrix $B_{ij}$.

$$B_{ij} = \begin{bmatrix} 0.573 & 0.503 & 0.492 & 0.508 & 0.514 & 0.363 & 0.492 & 0.503 & 0.492 & 0.447 & 0.601 & 0.523 & 0.492 & 0.514 & 0.532 \\ 0.655 & 0.704 & 0.655 & 0.609 & 0.686 & 0.690 & 0.655 & 0.646 & 0.655 & 0.716 & 0.526 & 0.628 & 0.655 & 0.514 & 0.621 \\ 0.492 & 0.503 & 0.573 & 0.609 & 0.514 & 0.614 & 0.573 & 0.574 & 0.573 & 0.537 & 0.601 & 0.576 & 0.573 & 0.686 & 0.576 \end{bmatrix}.$$

The dynamic weight matrix $\overline{w}$ of the evaluation index (obtained using the G2 method) was multiplied by matrix $B_{ij}$ (obtained using the TOPSIS method) to obtain the weighted comprehensive evaluation matrix $Z_{ij}$:

$Z_{ij} = B_{ij}\overline{w}$

$$= \begin{bmatrix} 0.033 & 0.032 & 0.055 & 0.038 & 0.014 & 0.017 & 0.022 & 0.019 & 0.02 & 0.051 & 0.091 & 0.022 & 0.027 & 0.024 & 0.046 \\ 0.037 & 0.045 & 0.073 & 0.045 & 0.019 & 0.033 & 0.03 & 0.024 & 0.027 & 0.082 & 0.079 & 0.026 & 0.036 & 0.024 & 0.054 \\ 0.028 & 0.032 & 0.063 & 0.045 & 0.014 & 0.029 & 0.026 & 0.022 & 0.023 & 0.061 & 0.091 & 0.024 & 0.031 & 0.032 & 0.05 \end{bmatrix}.$$

The weighted comprehensive evaluation matrix $Z_{ij}$ was substituted into Eq (17) to obtain the positive and negative ideal solutions of the evaluation objective:

$$f^{+} = (0.037 \quad 0.045 \quad 0.073 \quad 0.045 \quad 0.019 \quad 0.033 \quad 0.03 \quad 0.024 \quad 0.027 \quad 0.082 \quad 0.079 \quad 0.026 \quad 0.036 \quad 0.032 \quad 0.054),$$

$$f^{-} = (0.028 \quad 0.032 \quad 0.055 \quad 0.038 \quad 0.014 \quad 0.017 \quad 0.022 \quad 0.019 \quad 0.02 \quad 0.051 \quad 0.079 \quad 0.022 \quad 0.027 \quad 0.024 \quad 0.046).$$

Based on the obtained positive and negative ideal solutions, Eq (18) was used to calculate the Euclidean distance between the score data and the ideal value:

$$S_i^{+} = (0.048 \quad 0.008 \quad 0.018)^{T}; \quad S_i^{-} = (0.013 \quad 0.047 \quad 0.042)^{T}.$$

Based on the Euclidean distance, Eq (19) was used to calculate the relative closeness between

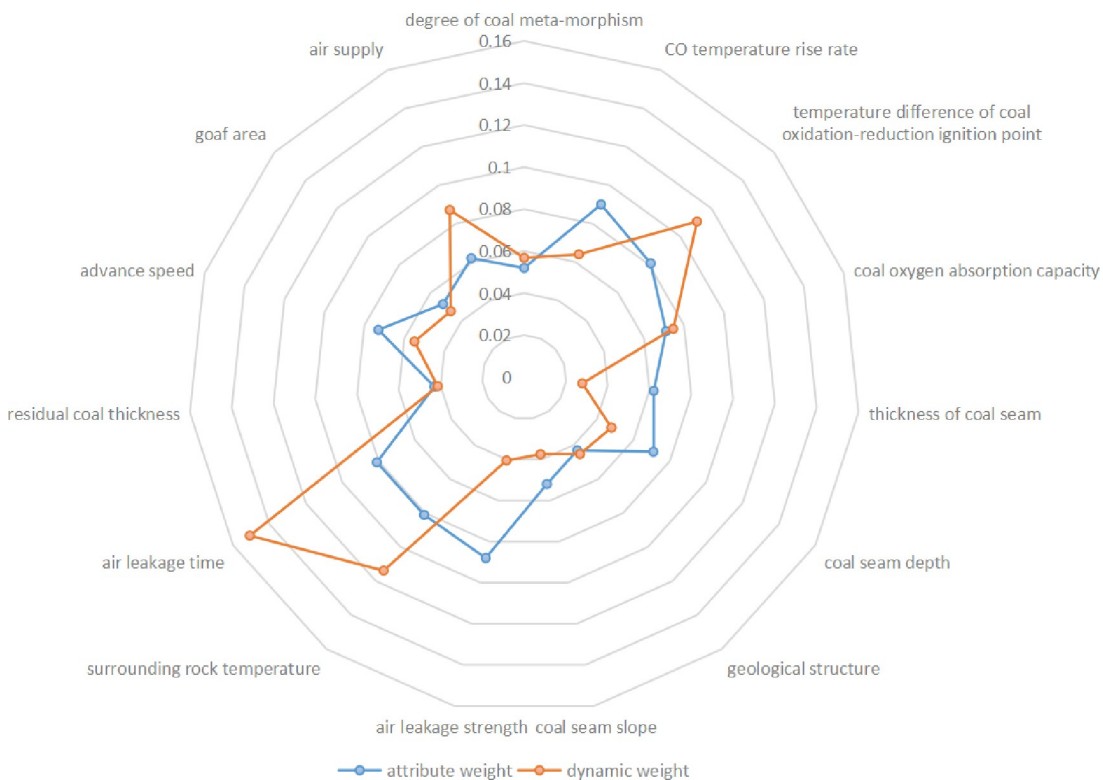

**Fig 2. Weight comparison radar chart.**

each score and each index in the criterion layer:

$$N_i^+ = \begin{pmatrix} 0.213 \\ 0.855 \\ 0.70 \end{pmatrix}.$$

According to the calculation results of the relative proximity, the second scoring has the largest relative proximity, which is 0.855. This shows that the secondary scoring is the most suitable dynamic prediction model for spontaneous goaf combustion, while the first scoring has a lower relative proximity. The reason for this is that in the early stage, the goaf does not experience the conditions for spontaneous combustion of the residual coal.

## 5.3 Use of the G2-TOPSIS dynamic model to evaluate the spontaneous goaf combustion risk level

According to the calculations of the comprehensive weights and the relative closeness, the second and third scoring judgement results are better. The second and third scoring results were evaluated using the G2-TOPSIS method. The dynamic weights $\overline{w}_D$ were multiplied by the evaluation matrix $V$, which is composed of expert scoring, to obtain the G2-TOPSIS dynamic evaluation result $L$.

According to the G2-TOPSIS dynamic evaluation model (Eq (20)), the result of the spontaneous goaf combustion risk assessment is $L = 69.349 \in (60, 70]$, and the predicted spontaneous goaf combustion risk level is III. This is consistent with the actual goaf spontaneous combustion situation, indicating that the model's prediction results are suitable for field applications.

## 6 Conclusions

1) Based on the improved CRITIC-G2-TOPSIS dynamic prediction model for spontaneous goaf combustion, the dynamic characteristics of the goaf and experts' experiences were comprehensively considered, overcoming the influences of the double defects, i.e., the variation degree of the index data and decision-maker's intentions. This avoids the problem that the combination coefficient cannot be scientifically and reasonably allocated, and the evaluation results are more authentic and credible.

2) Based on experience, knowledge, and preferences, the improved CRITIC and G2 methods were used to rank the importance of each index in the index information, and rational values were assigned to the confidence intervals of each index. Calculating the weight of each index not only eliminates the influence of the correlation coefficients with dimensional standard deviation and equal absolute values but also considers the data variation range of each index. Finally, the ideal comprehensive weight of each index was obtained, and the update factor was introduced to change the comprehensive weight into a dynamic weight, which is suitable for the dynamic environment.

3) The weights of the evaluation indexes obtained using the dynamic comprehensive weighting method show that the degree of influence of each index on spontaneous goaf combustion is different. The primary and secondary factors affecting the spontaneous goaf combustion risk of working face 1303 in the Jinniu Coal Mine are as follows: the air leakage time ($U'$ 33) > the temperature of the surrounding rock ($U'$ 32) > the temperature difference of the coal oxidation and reduction ignition point ($U'$ 13) > the air supply ($U'$ 44).

4) Considering the factors influencing spontaneous goaf combustion from a dynamic point of view, an evaluation index system for the spontaneous goaf combustion risk level was established based on the Delphi principle. Moreover, an effective method for predicting the spontaneous goaf combustion risk was developed. The spontaneous goaf combustion risk grade was predicted to be grade III, which is consistent with the actual situation in the field.

5) The improved CRITIC method, G2 method and TOPSIS method are combined to predict the risk of residual coal spontaneous combustion in goaf. However, there are some new intelligent algorithms. We should continue to study the optimization combination of these algorithms with this algorithm and apply them to the risk prediction of residual coal spontaneous combustion in goaf.

## Supporting information

**S1 File.**
(DOCX)

## Acknowledgments

The research presented in this paper was supported by the Key Laboratory of Mine Thermodynamic disasters and Control of Ministry of Education (Liaoning Technical University), Huludao, China. We thank LetPub (www.letpub.com) for its linguistic assistance during the preparation of this manuscript.

## Author Contributions

**Data curation:** Wei Wang.

**Funding acquisition:** Wei Wang, Baoshan Jia.

**Investigation:** Wei Wang.

**Methodology:** Wei Wang.

**Project administration:** Baoshan Jia, Youli Yao.

**Resources:** Wei Wang, Yun Qi, Youli Yao.

**Software:** Wei Wang, Yun Qi.

**Writing – original draft:** Wei Wang, Yun Qi.

**Writing – review & editing:** Wei Wang, Yun Qi.

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
