## [Decision Letter · Decision Letter 0]

30 Jul 2021

PONE-D-21-20758

Dynamic p rediction  model of spontaneous combustion risk  in goaf  based on improved CRITIC-G2-TOPSIS method and its application

PLOS ONE

Dear Dr. wang,

Thank you for submitting your manuscript to PLOS ONE. After careful consideration, we feel that it has merit but does not fully meet PLOS ONE’s publication criteria as it currently stands. Therefore, we invite you to submit a revised version of the manuscript that addresses the points raised during the review process.

ACADEMIC EDITOR:

1) Why you have used G2 method for determining criteria weights? Why not BWM, FUCOM or Level Based Weight Assessment (LBWA) methods? These methods should be discussed. The authors need to discuss their contributions compared to those in related papers.  The authors must clearly discuss the significance of the research problem in the first section.

2) How this modification of CRITIC method is better than the following modification https://doi.org/10.31181/dmame2003149z . This should be discussed in the manuscript.

We look forward to receiving your revised manuscript.

Kind regards,

Dragan Pamucar

Academic Editor

PLOS ONE

Journal Requirements:

“no”

6. Please amend the manuscript submission data (via Edit Submission) to include author Baoshan JIA and Youli YAO.

7. You noted that "Wei WANG, Yun QI, Baoshan JIA, Youli YAO" is a group author whereas these are the authors for this submission. Please confirm that there is indeed a group author for this submission. In addition to naming the author group, please list the individual authors and affiliations within this group in the acknowledgments section of your manuscript. Please also indicate clearly a lead author for this group along with a contact email address.

Reviewers' comments:

Reviewer's Responses to Questions

**Comments to the Author**

1. Is the manuscript technically sound, and do the data support the conclusions?

Reviewer #1: Yes

Reviewer #2: Yes

2. Has the statistical analysis been performed appropriately and rigorously? 

Reviewer #1: Yes

Reviewer #2: Yes

3. Have the authors made all data underlying the findings in their manuscript fully available?

Reviewer #1: Yes

Reviewer #2: Yes

4. Is the manuscript presented in an intelligible fashion and written in standard English?

Reviewer #1: Yes

Reviewer #2: Yes

5. Review Comments to the Author

Reviewer #1: Spontaneous combustion of residual coal in goaf is one of the main disasters that affect the safety production of coal mine. It often leads to serious coal resources damage and casualties. Under some special conditions, it can also induce gas explosion and other major malignant accidents. In this paper, considering the existing problems in the risk prediction of spontaneous combustion of residual coal in goaf during the dynamic advancing process of working face, aiming at the shortcomings of the previous established risk prediction model of spontaneous combustion in goaf, a dynamic prediction model of spontaneous combustion in goaf based on improved CRITIC-G2-TOPSIS is proposed, which is a relatively new method of coal spontaneous combustion disaster prediction. It has certain theoretical and engineering application value for revealing the spontaneous combustion characteristics of residual coal in goaf and preventing and controlling such accidents, however, the author still needs to pay attention to the following problems and suggestions:

1.The introduction of this paper can further elaborate the research progress of CRITIC method in the prediction of spontaneous combustion of residual coal in goaf.

2.How to determine the value range of evaluation grade of spontaneous combustion of gob residual coal in Section 3.2 of this paper?

3.In section 1.4 of the paper, "the determination of dynamic weight" is not clear about whether the 15 secondary indicators are used to determine the weight dynamically.

4.This paper studies the dynamic prediction of goaf spontaneous combustion risk based on improved CRITIC-G2-TOPSIS, how is it achieved?

5.In the conclusion of the paper, "3) The weights of the evaluation indexes obtained using the dynamic comprehensive weighting method show that the degree of influence of each index on spontaneous goaf combustion is different. The primary and secondary factors affecting the spontaneous goaf combustion risk of working face 1303 in the Jinniu Coal Mine are as follows: ...", which is best explained in the paper.

6.In Section 4.3 of this paper, the dynamic evaluation results of G2-TOPSIS are suggested to be transformed from vector product to column vector transpose or to give the calculation results directly.

7.There are a lot of articles using mathematical prediction model to evaluate the risk of spontaneous combustion in goaf. Where is the innovation of the paper?

8.By establishing a dynamic prediction model of goaf spontaneous combustion risk based on improved CRITIC-G2-TOPSIS, this paper makes a dynamic analysis on the spontaneous combustion risk of residual coal in 1303 working face of Jinniu coal mine. Why do you choose this model and what are its advantages? Please specify.

9.The latest references related to the content of the article are few. It is recommended to increase the number of references in the past three years.

10.Please modify the acknowledgment content of the paper. Acknowledgements are generally to thank the project sponsors and colleagues who have contributed to the experiment and writing of the paper but are not the author.

Reviewer #2: Thank you for inviting me as a reviewer for the paper titled Dynamic prediction model of spontaneous combustion risk in goaf based on improved CRITIC-G2-TOPSIS method and its application. This topic is important for investigation, and I am giving support to the authors. The paper has well structure. For the paper to be accepted, certain refinements need to be made:

- Literature analysis needs to be improved. Add another 10-15 papers of more recent date (period 2019-2021), such as:

o https://doi.org/10.31181/dmame2003149z

o https://doi.org/10.31181/rme200101162k

o doi.org/10.22190/FUME200305031A

- Compare the results with other methods (for example: VIKOR, MABAC, DEMATEL, etc.).

- Systematize the advantages and limitations of your research study.

- Add future research in conclusion.

6. PLOS authors have the option to publish the peer review history of their article (what does this mean?). If published, this will include your full peer review and any attached files.

Reviewer #1: No

Reviewer #2: No

---

## [Author Response · Author response to Decision Letter 0]

11 Aug 2021

Dear editor and reviewers:

Thank you for giving us the opportunity to submit a revised draft of the manuscript "Dynamic prediction model of spontaneous combustion risk in goaf based on improved CRITIC-G2-TOPSIS method and its application". We appreciate the time and effort that you and the reviewers dedicated to providing feedback on our manuscript and are grateful for the insightful comments and valuable improvements to our paper. 

We have incorporated the suggestions made by academic editor and reviewers. Those changes are highlighted in the manuscript. Please see below, in green, for a point-by-point response to the academic editor and reviewers’ comments.

Replies to the academic editor’s comments:

1) Why you have used G2 method for determining criteria weights? Why not BWM, FUCOM or Level Based Weight Assessment (LBWA) methods? These methods should be discussed. The authors need to discuss their contributions compared to those in related papers.  The authors must clearly discuss the significance of the research problem in the first section.

Response: in this paper, the G2 subjective weighting method based on "function driven" is mainly used, which is often called the unique reference comparison judgment method. G2 weighting method is a "function driven" subjective weighting method that quantifies the qualitative analysis process of information and risk uncertainty according to the relative importance of each evaluation index, and then calculates the weight of each evaluation index in group decision-making within a given score range. G2 method has obvious advantages, which can better reflect the knowledge, experience, information and risk awareness of experts. At the same time, G2 method has convenient modeling and simple calculation process, but this method also has shortcomings. For example, expert knowledge and experience have a great impact on defining the importance of indicators, As a result, the final index weight depends too much on the subjectivity of experts, which is easy to produce great differences. In order to avoid these problems, the improved critical method is integrated into the G2 method for coupling. With the help of the improved critical method, the importance ratio between indicators is modified, so that the subjective weighting results have both data objectivity and the effective combination of subjective and objective information. Compared with BWM, fucom or lbwa, it has obvious advantages. At the same time, due to the particularity of coal spontaneous combustion environment in goaf, the above method is not suitable for predicting the risk of coal spontaneous combustion in goaf. In addition, the significance of the research problem has been clearly discussed in the first section as required.

2) How this modification of CRITIC method is better than the following modification https://doi.org/10.31181/dmame2003149z . This should be discussed in the manuscript.

 Response: CRITIC method uses standard deviation to measure the difference degree of indicators, but the solution process of standard deviation is vulnerable to the extreme value of data, so it is necessary to eliminate the impact of extreme data on the analysis results. In order to reflect the degree of data dispersion more objectively and further improve the calculation accuracy of CRITIC method, it needs to be optimized and improved. The improved CRITIC method is used to calculate the index weight, which makes up for the defects caused by the differences in the order of magnitude and dimension between the indexes when the standard deviation is used to measure the discrimination in the CRITIC method.

This paper proposes an improved CRITIC weighting method, which is a "difference driven" objective weighting method, which introduces the mean difference of evaluation indexes according to the variation degree of each evaluation index and its influence on other indexes, and uses the ratio of information instead of the decision-maker's subjective determination of the relative importance ratio to determine the index weight. At the same time, the improved CRITIC method is integrated into the G2 method for coupling. With the help of the improved CRITIC method, the importance ratio between indicators is corrected, so that the subjective weighting results have both data objectivity and realize the effective combination of subjective and objective information. In addition, relevant discussions have been held in the manuscript as required.

Replies to the reviewers’ comments:

Response to reviewers 1

1. The introduction of this paper can further elaborate the research progress of CRITIC method in the prediction of spontaneous combustion of residual coal in goaf.

 Response: at present, there is no relevant research on the application of critical method to the prediction of residual coal spontaneous combustion in goaf. The introduction has been modified according to the modification requirements, and the relevant literature in recent three years has been added.

2. How to determine the value range of evaluation grade of spontaneous combustion of gob residual coal in Section 3.2 of this paper?

 Response: the classification and assignment of spontaneous combustion risk level are determined according to relevant regulations, and relevant literature has been cited.

3. In section 1.4 of the paper, "the determination of dynamic weight" is not clear about whether the 15 secondary indicators are used to determine the weight dynamically.

 Response: the determination of dynamic weight in this paper is to dynamically determine the weight of 15 secondary indicators, which has been explained in this paper according to the modification requirements.

4. This paper studies the dynamic prediction of goaf spontaneous combustion risk based on improved CRITIC-G2-TOPSIS, how is it achieved?

 Response: with the continuous advancement of coal mining face, the natural ignition environment of residual coal in goaf has changed, so the risk of spontaneous combustion in goaf also changes dynamically. The previous prediction and evaluation model of spontaneous combustion risk in goaf can not well reflect the dynamic development of residual coal spontaneous combustion in goaf, The dynamic prediction method of goaf spontaneous combustion risk proposed in this paper is an improvement of the fixed weight in the traditional evaluation method. It is based on the coupling of the comprehensive weight of the improved critical correction G2 method and the update factor of each index. It can correct the weight according to the feedback information of each index with the change of goaf environment in the evaluation system, and timely adjust the impact of each index on the risk of goaf spontaneous combustion, improve the flexibility of the evaluation system.

5. In the conclusion of the paper, "3) The weights of the evaluation indexes obtained using the dynamic comprehensive weighting method show that the degree of influence of each index on spontaneous goaf combustion is different. The primary and secondary factors affecting the spontaneous goaf combustion risk of working face 1303 in the Jinniu Coal Mine are as follows: ...", which is best explained in the paper.

 Response: it has been supplemented in the corresponding chapters of the text in accordance with relevant modification suggestions.

6. In Section 4.3 of this paper, the dynamic evaluation results of G2-TOPSIS are suggested to be transformed from vector product to column vector transpose or to give the calculation results directly.

 Response: the vector product has been modified as required.

7. There are a lot of articles using mathematical prediction model to evaluate the risk of spontaneous combustion in goaf. Where is the innovation of the paper?

 Response: in this paper, an improved critical information modified G2 weighting method is proposed. On the basis of taking into account the variability and conflict of index data, the disadvantage that a single weighting method can not reflect subjective and objective information at the same time is avoided. The g2-topsis mathematical prediction model is established according to the superior superior solution distance method (TOPSIS), and applied to the research field of residual coal spontaneous combustion in coal mine goaf, A dynamic mathematical prediction model is established for the first time to study the dynamic change of spontaneous combustion risk in goaf during the advancement of working face.

8. By establishing a dynamic prediction model of goaf spontaneous combustion risk based on improved CRITIC-G2-TOPSIS, this paper makes a dynamic analysis on the spontaneous combustion risk of residual coal in 1303 working face of Jinniu coal mine. Why do you choose this model and what are its advantages? Please specify.

 Response: based on the improved CRITIC-G2-TOPSIS dynamic prediction model for spontaneous goaf combustion, the dynamic characteristics of the goaf and experts’ experiences were comprehensively considered, overcoming the influences of the double defects, i.e., the variation degree of the index data and decision-maker's intentions. This avoids the problem that the combination coefficient cannot be scientifically and reasonably allocated, and the evaluation results are more authentic and credible.

9. The latest references related to the content of the article are few. It is recommended to increase the number of references in the past three years.

 Response: relevant references for nearly three years have been added as required.

10. Please modify the acknowledgment content of the paper. Acknowledgments are generally to thank the project sponsors and colleagues who have contributed to the experiment and writing of the paper but are not the author.

Response: the acknowledgment content of the paper has been modified according to the modification requirements.

Response to reviewers 2

1. Literature analysis needs to be improved. Add another 10-15 papers of more recent date (period 2019-2021), such as:

o https://doi.org/10.31181/dmame2003149z

o https://doi.org/10.31181/rme200101162k

o doi.org/10.22190/FUME200305031A

Response: the literature analysis has been improved as required, and 10 recent papers similar to the research direction of this paper have been added.

2. Compare the results with other methods (for example: VIKOR, MABAC, DEMATEL, etc.).

 Response: it has been modified as required.

3. Systematize the advantages and limitations of your research study.

Response: it has been modified as required.

4. Add future research in conclusion.

Response: future research has been added to the conclusion (the replies are highlighted in green in the paper). 

Once again, thank you very much for your constructive comments and suggestions which would help us both in English and in depth to improve the quality of the paper.

Kind regards,

Wei WANG

E-mail: wangwei@sxdtdx.edu.cn,

Corresponding author: Yun QI,

E-mail address: qiyun_sx@sxdtdx.edu.cn .

---

## [Decision Letter · Decision Letter 1]

3 Sep 2021

Dynamic p rediction  model of spontaneous combustion risk  in goaf  based on improved CRITIC-G2-TOPSIS method and its application

PONE-D-21-20758R1

Dear Dr. wang,

We’re pleased to inform you that your manuscript has been judged scientifically suitable for publication and will be formally accepted for publication once it meets all outstanding technical requirements.

Kind regards,

Dragan Pamucar

Academic Editor

PLOS ONE

Additional Editor Comments (optional):

Reviewers' comments:

Reviewer's Responses to Questions

**Comments to the Author**

1. If the authors have adequately addressed your comments raised in a previous round of review and you feel that this manuscript is now acceptable for publication, you may indicate that here to bypass the “Comments to the Author” section, enter your conflict of interest statement in the “Confidential to Editor” section, and submit your "Accept" recommendation.

Reviewer #1: All comments have been addressed

Reviewer #2: All comments have been addressed

2. Is the manuscript technically sound, and do the data support the conclusions?

Reviewer #1: Yes

Reviewer #2: Yes

3. Has the statistical analysis been performed appropriately and rigorously? 

Reviewer #1: Yes

Reviewer #2: Yes

4. Have the authors made all data underlying the findings in their manuscript fully available?

Reviewer #1: Yes

Reviewer #2: Yes

5. Is the manuscript presented in an intelligible fashion and written in standard English?

Reviewer #1: Yes

Reviewer #2: Yes

6. Review Comments to the Author

Reviewer #1: (No Response)

Reviewer #2: All the reviewers' comments have been addressed carefully and sufficiently. The revisions are rational from my point of view. I think the current version of the paper can be accepted.

7. PLOS authors have the option to publish the peer review history of their article (what does this mean?). If published, this will include your full peer review and any attached files.

Reviewer #1: No

Reviewer #2: No

---

## [Editor Report · Acceptance letter]

14 Oct 2021

PONE-D-21-20758R1 

Dynamic prediction  model of spontaneous combustion risk  in goaf  based on improved CRITIC-G2-TOPSIS method and its application 

Dear Dr. wang:

I'm pleased to inform you that your manuscript has been deemed suitable for publication in PLOS ONE. Congratulations! Your manuscript is now with our production department. 

Kind regards, 

on behalf of

Dr. Dragan Pamucar 

Academic Editor

PLOS ONE